# A Powerful Generative Model Using Random Weights for the Deep Image Representation

**Kun He**,* **Yan Wang** [†]
Department of Computer Science and Technology
Huazhong University of Science and Technology, Wuhan 430074, China
brooklet60@hust.edu.cn, yanwang@hust.edu.cn

**John Hopcroft**
Department of Computer Science
Cornell University, Ithaca 14850, NY, USA
jeh@cs.cornell.edu

## Abstract

To what extent is the success of deep visualization due to the training? Could we do deep visualization using untrained, random weight networks? To address this issue, we explore new and powerful generative models for three popular deep visualization tasks using untrained, random weight convolutional neural networks. First we invert representations in feature spaces and reconstruct images from white noise inputs. The reconstruction quality is statistically higher than that of the same method applied on well trained networks with the same architecture. Next we synthesize textures using scaled correlations of representations in multiple layers and our results are almost indistinguishable with the original natural texture and the synthesized textures based on the trained network. Third, by recasting the content of an image in the style of various artworks, we create artistic images with high perceptual quality, highly competitive to the prior work of Gatys et al. on pretrained networks. To our knowledge this is the first demonstration of image representations using untrained deep neural networks. Our work provides a new and fascinating tool to study the representation of deep network architecture and sheds light on new understandings on deep visualization. It may possibly lead to a way to compare network architectures without training.

## 1   Introduction

In recent years, Deep Neural Networks (DNNs), especially Convolutional Neural Networks (CNNs), have demonstrated highly competitive results on object recognition and image classification [1, 2, 3, 4]. With advances in training, there is a growing trend towards understanding the inner working of these deep networks. By training on a very large image data set, DNNs develop a representation of images that makes object information increasingly explicit at various levels of the hierarchical architecture. Significant visualization techniques have been developed to understand the deep image representations on trained networks [5, 6, 7, 8, 9, 10, 11].

Inversion techniques have been developed to create synthetic images with feature representations similar to the representations of an original image in one or several layers of the network. Feature representations are a function $\Phi$ of the source image $x_0$. An approximate inverse $\Phi^{-1}$ is used to

[†]Corresponding author.

construct a new image $x$ from the code $\Phi(x_0)$ by reducing some statistical discrepancy between $\Phi(x)$ and $\Phi(x_0)$. Mahendran et al. [7] use the pretrained CNN AlexNet [2] and define a squared Euclidean loss on the activations to capture the representation differences and reconstruct the image. Gatys et al. [8, 12] define a squared loss on the correlations between feature maps of some layers and synthesize natural textures of high perceptual quality using the pretrained CNN called VGG [3]. Gatys et al. [13] then combine the loss on the correlations as a proxy to the style of a painting and the loss on the activations to represent the content of an image, and successfully create artistic images by converting the artistic style to the content image, inspiring several followups [14, 15]. Another stream of visualization aims to understand what each neuron has learned in a pretrained network and synthesize an image that maximally activates individual features [5, 9] or the class prediction scores [6]. Nguyen et al. further try multifaceted visualization to separate and visualize different features that a neuron learns [16].

Feature inversion and neural activation maximization both start from a white noise image and calculate the gradient via backpropagation to morph the white noise image and output a natural image. In addition, some regularizers are incorporated as a natural image prior to improve the visualization quality, including $\alpha-$norm [6], total variation [7], jitter [7], Gaussian blur [9], data-driven patch priors [17], etc. The method of visualizing the feature representation on the intermediate layers sheds light on the information represented at each layer of the pretrained CNN.

A third set of researchers trains a separate feed-forward CNN with deconvolutional layers using representations or correlations of the feature maps produced in the original network as the input and the source image as the target to learn the inversion of the original network. The philosophy is to train another neural network to inverse the representation and speedup the visualization on image reconstruction [10, 18], texture synthesis [19] or even style transfer [15]. Instead of designing a natural prior, some researchers incorporate adversarial training [20] to improve the realism of the generated images [18]. Their trained deconvolutional network could give similar qualitative results as the inversion technique does and is two or three orders of magnitude faster, as the previous inversion technique needs a forward and backward pass through the pretrained network. This technique is slightly different from the previous two in that it does not focus on understanding representations encoded in the original CNN but on the visualization of original images by training another network.

It is well recognized that deep visualization techniques conduct a direct analysis of the visual information contained in image representations, and help us understand the representation encoded at the intermediate layers of the well trained DNNs. In this paper, we raise a fundamental issue that other researchers rarely address: **Could we do deep visualization using untrained, random weight DNNs? What kind of deep visualization could be applied on random weight DNNs? This would allow us to separate the contribution of training from the contribution of the network structure. It might even give us a method to evaluate deep network architectures without spending days and significant computing resources in training networks so that we could compare them. Also, it will be useful not to have to store the weights, which can have significant impact for mobile applications.** Though Gray et al. demonstrated that the VGG architecture with random weights failed in generating textures and resulted in white noise images in an experiment indicating the trained filters might be crucial for texture generation [8], we conjecture the success of deep visualization mainly originates from the intrinsic nonlinearity and complexity of the deep network hierarchical structure rather than from the training, and that the architecture itself may cause the inversion invariant to the original image. Gatys et al.'s unsuccessful attempt on the texture synthesis using the VGG architecture with random weights may be due to their inappropriate scale of the weighting factors.

To verify our hypothesis, we try three popular inversion tasks for visualization using the CNN architecture with random weights. Our results strongly suggest that this is true. Applying inversion techniques on the *untrained* VGG with random weights, we reconstruct high perceptual quality images. The results are qualitatively better than the reconstructed images produced on the pretrained VGG with the same architecture. Then, we try to synthesize natural textures using the random weight VGG. With automatic normalization to scale the squared correlation loss for different activation layers, we succeed in generating similar textures as the prior work of Gatys et al. [8] on well-trained VGG. Furthermore, we continue the experiments on style transfer, combining the content of an image and the style of an artwork, and create artistic imagery using random weight CNN.

To our knowledge this is the first demonstration of image representations using untrained deep neural networks. Our work provides a new and fascinating tool to study the perception and representation of deep network architecture, and shed light on new understandings on deep visualization. Our work will inspire more possibilities of using the generative power of CNNs with random weights, which do not need long training time on multi-GPUs. Furthermore, it is very hard to prove why trained deep neural networks work so well. Based on the networks with random weights, we might be able to prove some properties of the deep networks. Our work using random weights shows a possible way to start developing a theory of deep learning since with well-trained weights, theorems might be impossible.

## 2 Methods

In order to better understand the deep representation in the CNN architecture, we focus on three tasks: inverting the image representation, synthesizing texture, and creating artistic style images. Our methods are similar in spirit to existing methods [7, 8, 13]. The main difference is that we use untrained weights instead of trained weights, and we apply weighting factors determined by a pre-process to normalize the different impact scales of different activation layers on the input layer. Compared with purely random weight CNN, we select a random weight CNN among a set of random weight CNNs to get slightly better results.

For the reference network, we choose VGG-19 [3], a convolutional neural network trained on the 1.3 million-image ILSVRC 2012 ImageNet dataset [1] using the Caffe-framework [22]. The VGG architecture has 16 convolutional and 5 pooling layers, followed by 3 fully connected layers. Gatys et al. re-train the VGG-19 network using average pooling instead of maximum pooling, which they suggest could improve the gradient flow and obtain slightly better results [8]. They only consider the convolutional and pooling layers for texture synthesis, and they rescale the weights such that the mean activation of each filter over the images and positions is 1. Their trained network is denoted as *VGG* in the following discussion. We adopt the same architecture, replacing the weights with purely random values from a Gaussian distribution $N(0, \sigma)$. The standard deviation, $\sigma$, is set to a small number like 0.015 in the experiments. The VGG-based random weight network created as described in the following subsection is used as our reference network, denoted as *ranVGG* in the following discussion.

**Inverting deep representations.** Given a representation function $F^l : \mathbb{R}^{H \times W \times C} \rightarrow \mathbb{R}^{N_l \times M_l}$ for the $l^{th}$ layer of a deep network and $F^l(\mathbf{x}_0)$ for an input image $\mathbf{x}_0$, we want to reconstruct an image $\mathbf{x}$ that minimizes the $L_2$ loss among the representations of $\mathbf{x}_0$ and $\mathbf{x}$.

$$\mathbf{x}^* = \operatorname*{argmin}_{\mathbf{x} \in \mathbb{R}^{H \times W \times C}} \mathcal{L}_{content}(\mathbf{x}, \mathbf{x}_0, l) = \operatorname*{argmin}_{\mathbf{x} \in \mathbb{R}^{H \times W \times C}} \frac{\omega_l}{2 N_l M_l} \| F^l(\mathbf{x}) - F^l(\mathbf{x}_0) \|_2^2 \tag{1}$$

Here $H$ and $W$ denote the size of the image, $C = 3$ the color channels, and $\omega_l$ the weighting factor. We regard the feature map matrix $F^l$ as the representation function of the $l^{th}$ layer which has $N_l \times M_l$ dimensions where $N_l$ is the number of distinct feature maps, each of size $M_l$ when vectorised. $F^l_{ik}$ denotes the activation of the $i^{th}$ filter at position $k$.

The representations are a chain of non-linear filter banks even if untrained random weights are applied to the network. We initialize the pre_image with white noise, and apply the L_BFGS gradient descent using standard error backpropagation to morph the input pre_image to the target.

$$\mathbf{x}_{t+1} = \mathbf{x}_t - \left( \frac{\partial \mathcal{L}(\mathbf{x}, \mathbf{x}_0, l)}{\partial F^l} \frac{\partial F^l}{\partial \mathbf{x}} \right) \Bigg|_{\mathbf{x}_t} \tag{2}$$

$$\frac{\partial \mathcal{L}(\mathbf{x}, \mathbf{x}_0, l)}{\partial F^l_{i,k}} \Bigg|_{\mathbf{x}_t} = \frac{\omega_l}{N^l M^l} (F^l(\mathbf{x}_t) - F^l(\mathbf{x}_0))_{i,k} \tag{3}$$

The weighting factor $\omega_l$ is applied to normalize the gradient impact on the morphing image $\mathbf{x}$. We use a pre-processing procedure to determine the value of $\omega_l$. For the current layer $l$, we approximately calculate the maximum possible gradient by Equation (4), and back propagate the gradient to the input layer. Then we regard the reciprocal of the absolute mean gradient over all pixels and RGB channels as the value of $\omega_l$ such that the gradient impact of different layers is approximately of the same scale. This normalization doesn't affect the reconstruction from the activations of a single layer,

but is added for the combination of content and style for the style transfer task.

$$\frac{1}{\omega_l} = \frac{1}{WHC} \left| \sum_{i=1}^{W} \sum_{j=1}^{H} \sum_{k=1}^{C} \frac{\partial \mathcal{L}(\mathbf{x}_0, \mathbf{x}', l)}{\partial \mathbf{x}_{i,j,k}} \right|_{F^l(\mathbf{x}')=0} \tag{4}$$

To stabilize the reconstruction quality, we apply a greedy approach to build a "stacked" random weight network *ranVGG* based on the VGG-19 architecture. Select one single image as the reference image and starting from the first convolutional layer, we build the stacked random weight VGG by sampling, selecting and fixing the weights of each layer in forward order. For the current layer $l$, fix the weights of the previous $l-1$ layers and sample several sets of random weights connecting the $l^{th}$ layer. Then reconstruct the target image using the rectified representation of layer $l$, and choose weights yielding the smallest loss. Experiments in the next section show our success on the reconstruction by using the untrained, random weight CNN, *ranVGG*.

**Texture synthesis.** Can we synthesize natural textures based on the feature space of an untrained deep network? To address this issue, we refer to the method proposed by Gatys et al.[8] and use the correlations between feature responses on each layer as the texture representation. The inner product between pairwise feature maps $i$ and $j$ within each layer $l$, $G_{ij}^l = \sum_k F_{ik}^l F_{jk}^l$, defines a gram matrix $G^l = F^l(F^l)^T$. We seek a texture image $\mathbf{x}$ that minimizes the $L_2$ loss among the correlations of the representations of several candidate layers for $\mathbf{x}$ and a groundtruth image $\mathbf{x}_0$.

$$\mathbf{x}^* = \operatorname*{argmin}_{\mathbf{x} \in \mathbb{R}^{H \times W \times C}} \mathcal{L}_{texture} = \operatorname*{argmin}_{\mathbf{x} \in \mathbb{R}^{H \times W \times C}} \sum_{l \in L} \mu_l E(\mathbf{x}, \mathbf{x}_0, l), \tag{5}$$

where the contribution of layer $l$ to the total loss is defined as

$$E(\mathbf{x}, \mathbf{x}_0, l) = \frac{1}{4N_l^2 M_l^2} \| G^l(F^l(\mathbf{x})) - G^l(F^l(\mathbf{x}_0)) \|_2^2. \tag{6}$$

The derivative of $E(\mathbf{x}, \mathbf{x}_0, l)$ with respect to the activations $F^l$ in layer $l$ is [8]:

$$\frac{\partial E(\mathbf{x}, \mathbf{x}_0, l)}{\partial F_{i,k}^l} = \frac{1}{N_l^2 M_l^2} \{ (F^l(\mathbf{x}))^T [G^l(F^l(\mathbf{x})) - G^l(F^l(\mathbf{x}_0))] \}_{i,k} \tag{7}$$

The weighting factor $\mu_l$ is defined similarly to $\omega_l$, but here we use the loss contribution $E(\mathbf{x}, \mathbf{x}_0, l)$ of layer $l$ to get its gradient impact on the input layer.

$$\frac{1}{\mu_l} = \frac{1}{WHC} \left| \sum_i^W \sum_j^H \sum_k^C \frac{\partial E(\mathbf{x}_0, \mathbf{x}', l)}{\partial \mathbf{x}_{i,j,k}} \right|_{F^l(\mathbf{x}')=0} \tag{8}$$

We then perform the L_BFGS gradient descent using standard error backpropagation to morph the input image to a synthesized texture image using the untrained *ranVGG*.

**Style transfer.** Can we use the untrained deep network to create artistic images? Referring to the prior work of Gatys et al.[13] from the feature responses of VGG trained on ImageNet, we use an untrained VGG and succeed in separating and recombining content and style of arbitrary images. The objective requires terms for content and style respectively with suitable combination factors. For content we use the method of reconstruction on medium layer representations, and for style we use the method of synthesising texture on some lower through higher layer representation correlations.

Let $\mathbf{x}_c$ be the content image and $\mathbf{x}_s$ the style image. We combine the content of the former and the style of the latter by optimizing the following objective:

$$\mathbf{x}^* = \operatorname*{argmin}_{\mathbf{x} \in \mathbb{R}^{H \times W \times C}} \alpha \mathcal{L}_{content}(\mathbf{x}, \mathbf{x}_c) + \beta \mathcal{L}_{texture}(\mathbf{x}, \mathbf{x}_s) + \gamma \mathcal{R}(\mathbf{x}) \tag{9}$$

Here $\alpha$ and $\beta$ are the contributing factors for content and style respectively. We apply a regularizer $\mathcal{R}(\mathbf{x})$, total variation(TV) [7] defined as the squared sum on the adjacent pixel's difference of $\mathbf{x}$, to encourage the spatial smoothness in the output image.

## 3 Experiments

This section evaluates the results obtained by our model using the untrained network *ranVGG* [3].

The input image is required to be of size $256 \times 256$ if we want to invert the representation of the fully connected layers. Else, the input could be of arbitrary size.

**Inverting deep representations.**    We select several source images from the ILSVRC 2012 challenge [1] validation data as examples for the inversion task, and choose a monkey image as the reference image to build the stacked *ranVGG* (Note that using other image as the reference image also returns similar results). As compared with the inverting technique of Mahendran et al. [7], we only consider the Euclidean loss over the activations and ignore the regularizer they used to capture the natural image prior. *ranVGG* contains 19 layers of random weights (16 convolutional layers and 3 fully connected layers), plus 5 pooling layers. Mahendran et al. use a reference network AlexNet [2] which contains 8 layers of trained weights (5 convolutional layers and 3 fully connected layers), plus 3 pooling layers.

Figure 1 shows that we reach higher perceptive reconstructions. The reason may lie in the fact that the VGG architecture uses filters with a small receptive field of $3 \times 3$ and we adopt average pooling. Though shallower than VGG, their reference network, AlexNet, adopts larger filters and uses maximum pooling, which makes it harder to get images well inverted and easily leads to spikes. That's why they used regularizers to polish the reconstructed image. Figure 2 shows more examples (house, flamingo, girl).

Figure 3 shows the variations on an example image, the girl. As compared with the VGG with purely random weights, ranVGG (the VGG with stacked random weights) exhibits lower variations and lower reconstruction distances. As compared with the trained VGG, both stacked ranVGG and VGG with purely random weights exhibit lower reconstruction distance with lower variations. ranVGG demonstrates a more stable and high performance for the inversion task and is slightly better than an purely random VGG. So we will use ranVGG for the following experiments.

To compare the convergence of ranVGG and VGG, Figure 4 shows the loss (average Euclidean distance) along the gradient descent iterations on an example image, the house. The reconstruction converges much quicker on ranVGG and yields higher perceptual quality results. Note that the reconstruction on VGG remains the same even if we double the iteration limits to 4000 iterations.

**Texture synthesis.**    Figure 5 shows the textures synthesized by our model on ranVGG for several natural texture images (fifth row) selected from a texture website[4] and an artwork named *Starry Night* by Vincent van Gohn 1989. Each row of images was generated using an increasing number of convolutional layers to constrain the gradient descent. conv1_1 for the first row, conv1_1 and conv2_1 for the second row, etc (the labels at each row indicate the top-most layer included). The joint matching of conv1_1, conv2_1, and con3_1 (third row) already exhibits high quality texture representations. Adding one more layer of conv4_1 (fourth row) could slightly improve the natural textures. By comparison, results of Gatys et al.[8] on the trained VGG using four convolutional layers up to conv4_1 are as shown at the bottom row.

Our experiments show that with suitable weighted factors, calculated automatically by our method, ranVGG could synthesize complex natural textures that are almost indistinguishable with the original texture and the synthesized texture on the trained VGG. Trained VGG generates slightly better textures on neatly arranged original textures (cargo at the second column of Figure 5).

**Style transfer.**    We select conv2_2 as the content layer, and use the combination of conv1_1, conv2_1, ..., conv5_1 as the style. We set the ratio of $\alpha : \beta : \gamma = 100 : 1 : 1000$ in the experiments. We first compare our style transfer results with the prior work of Gatys et al.[13] on several well-known artworks for the style: *Starry Night* by Vincent van Gohn 1989, *Der Schrei* by Edward Munch 1893, *Picasso* by Pablo Picasso 1907, *Woman with a Hat* by Henri Matisse 1905, *Meadow with Poplars* by Claude Monet 1875. As shown in Figure 6, the second row, by recasting the content of a university image in the style of the five artworks, we obtain different artistic images based on the untrained ranVGG (second row). Our results are comparable to their work [13] on the pretrained VGG (third row), and are in the same order of magnitude. They have slightly smoother lines and textures which may attributed to the training. We further try the content and style combination on some Chinese paintings and scenery photographs, as shown in Figure 7, and create high perceptual artistic Chinese paintings that well combine the style of the painting and the content of the sceneries.

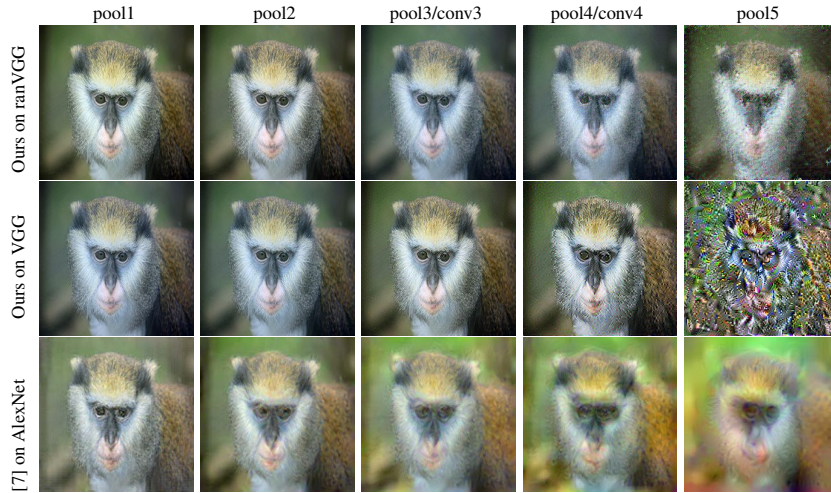

Figure 1: **Reconstructions from layers of ranVGG (top) and the pretrained VGG (middle) and [7] (bottom).** As AlexNet only contains 3 pooling layers, we compare their results on conv3 and conv4 with ours on pool3 and pool4. Our method on ranVGG demonstrates a higher perceptive quality, especially on the higher layers. Note that VGG is much deeper than AlexNet even when we compare on the same pooling layer.

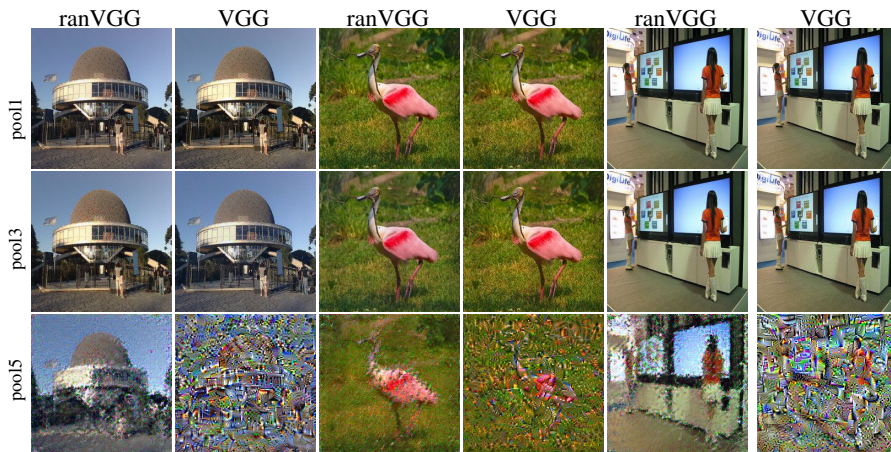

Figure 2: **Reconstructions from different pooling layers of the untrained ranVGG and the pretrained VGG.** ranVGG demonstrates a higher perceptive quality, especially on the higher layers. The pretrained VGG could rarely reconstruct even the contours from representations of the fifth pooling layer.

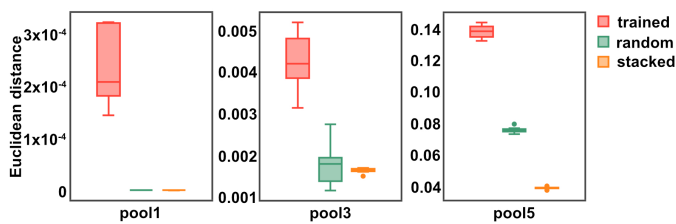

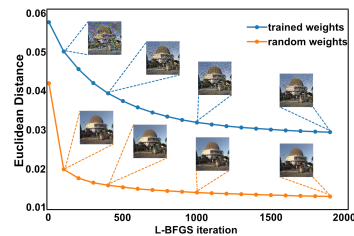

Figure 3: Variations in samples on the girl image, with maximum, minimum, mean and quartiles.

Figure 4: Reconstruction qualities of conv5_1 during the gradient descent iterations.

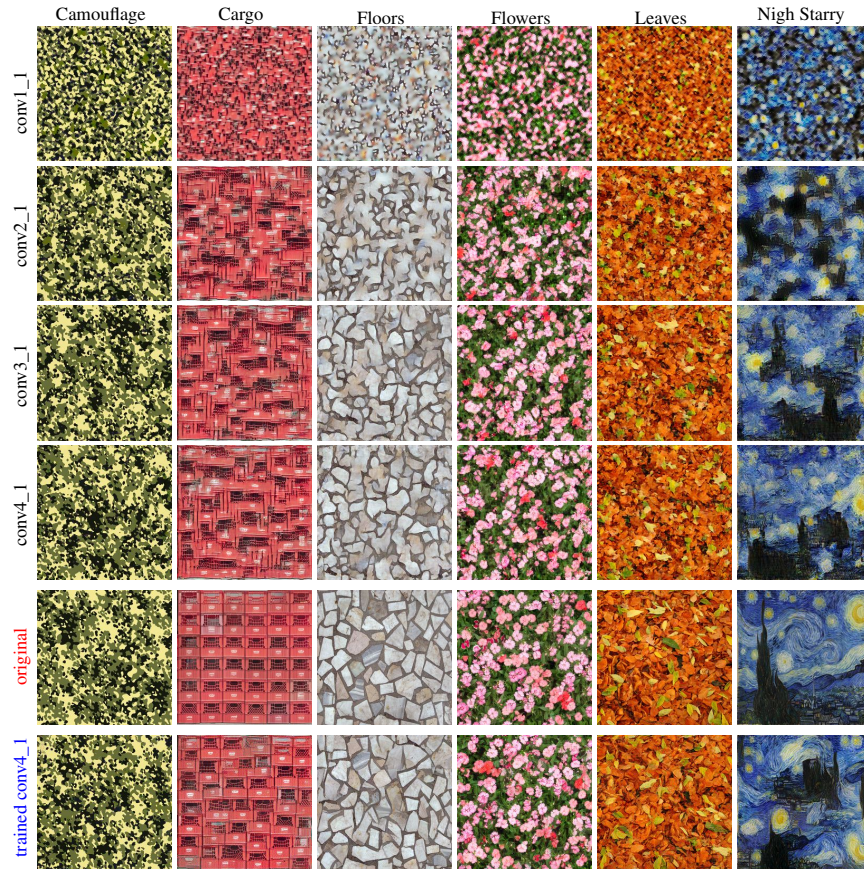

Figure 5: **Generated textures using random weights.** Each row corresponds to a different processing stage in ranVGG. Considering only the lowest layer, conv1_1, the synthesised textures are of lowest granularity, showing very local structure. Increasing the number of layers on which we match the texture representation (conv1_1 plus conv2_1 for the second row, etc), we have higher organizations of the previous local structure. The third row and the fourth row show high-quality synthesized textures of the original images. The lowest row corresponds to the result of using the trained VGG to match the texture representation from conv1_1, conv2_1 conv3_1 and conv4_1.

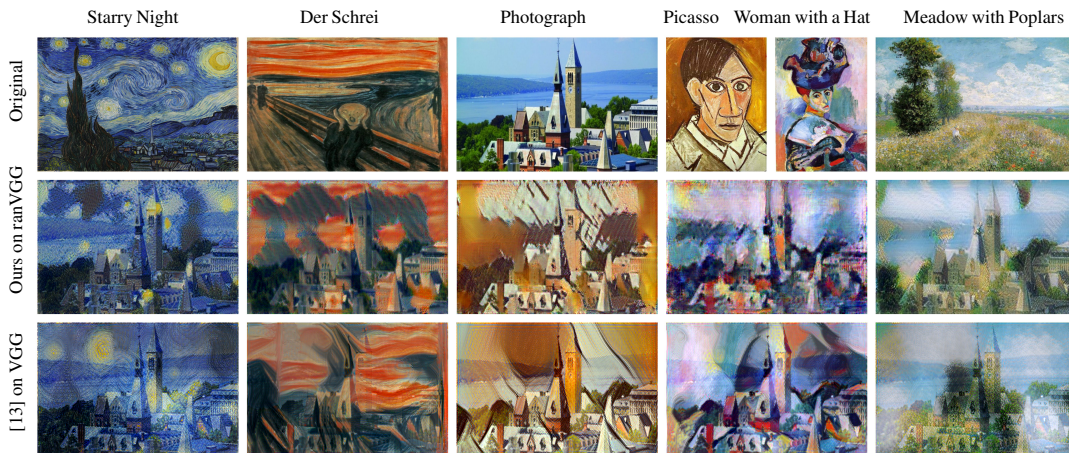

Figure 6: **Artistic style images of ours on the untrained ranVGG (medium row) and of Gatys et al.[8] on the pretrained VGG (bottom row).** We select a university image (first row, center) and several well-known artworks for the style (first row, others images). The third column under the photograph are for the Picasso. We obtain similar quality results as compared with Gatys et al.[13].

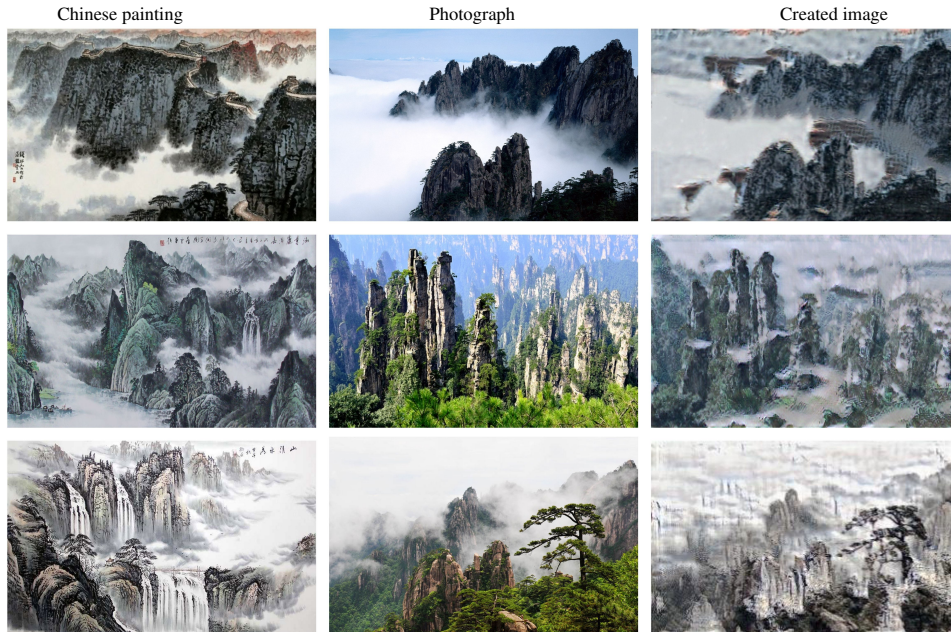

| Chinese painting | Photograph | Created image |

Figure 7: **Style transfer of Chinese paintings on the untrained ranVGG.** We select several Chinese paintings for the style (first column), including *The Great Wall* by Songyan Qian 1975, a painting of anonymous author and *Beautiful landscape* by Ping Yang. We select the mountain photographs (second column) as the content images. The created images performed on the untrained ranVGG are shown in the third column, which seem to have learned how to paint the rocks and clouds from paintings of the first column and transfer the style to the content to "draw" Chinese landscape paintings.

## 4   Discussion

Our work offers a testable hypothesis about the representation of image appearance based only on the network structure. The success on the untrained, random weight networks on deep visualization raises several fundamental questions in the area of deep learning. Researchers have developed many visualization techniques to understand the representation of well trained deep networks. However, if we could do the same or similar visualization using an untrained network, then the understanding is not for the training but for the network architecture. What is the difference of a trained network and a random weight network with the same architecture, and how could we explore the difference? What else could one do using the generative power of untrained, random weight networks? Explore other visualization tasks in computer vision developed on the well-trained network, such as image morphing [23], would be a promising aspect.

Training deep neural networks not only requires a long time but also significant high performance computing resources. The VGG network, which contains 11-19 weight layers depending on the typical architecture [3], takes 2 to 3 weeks on a system equipped with 4 NVIDIA Titan Black GPUs for training a single net. The residual network ResNet, which achieved state-of-the-art results in image classification and detection in 2015 [4], takes 3.5 days for the 18-layer model and 14 days for the 101-layer model using 4 NVIDIA Kepler GPU.[5] Could we evaluate a network structure without taking a long time to train it? There are some prior works to deal with this issue but they deal with much shallow networks [21]. In future work, we will address this issue by utilizing the untrained network to attempt to compare networks quickly without having to train them.

## Acknowledgments

This research work was supported by US Army Research Office(W911NF-14-1-0477) and National Science Foundation of China(61472147) and National Science Foundation of Hubei Province(2015CFB566).

## Footnotes

*The three authors contributing equally.

[3] https://github.com/mileyan/random_weights

[4]http://www.textures.com/

[5]http://torch.ch/blog/2016/02/04/resnets.html

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
