[Supplementary Material]

**Supplementary Information**

Figure 7: Architecture of the VGG network.

Figure 8: **Reconstructions of the monkey from each layer of the random weight CNN, ranVGG.**
The monkey image is well reconstructed from activations of any of the 16 convolutional layers after
the rectifier and the 5 average pooling layers, and we could still see the rough contours from the first
two fully connected layers.

287 Figure 9 shows the variations on one example image (the girl image at Figure 2). As compared with
288 the VGG with purely random weights, ranVGG, the VGG with stacked random weights, exhibits
289 lower variations and lower reconstruction distances. As compared with the trained VGG, both stacked
290 ranVGG and VGG with purely random weights exhibit lower reconstruction distance with lower
291 variations. ranVGG demonstrates a more stable and high performance for the inversion task.

Figure 9: Variations in samples on the girl image, with maxi-
mum, minimum, mean and quartiles.

Figure 10: Mean Euclidean dis-
tances on ranVGG.