[Reviews · NeurIPS 2016]

Reviewer 1

Summary

This paper demonstrate that style transfer and visualization can be accomplished from a randomly initialized CNN like the VGG network. This demonstrate the the feature matching is enabled largely by the CNN architecture rather than the learned network parameters.

Qualitative Assessment

I think this paper is very neat in demonstrating how random weights may be feasible to accomplish what the neural art methods have demonstrated. Over all this is a good paper which demonstrated something that is counter intuitive as to why style transfer worked well with deep CNNs. Figure 6 is also quite impressive in its qualitative results. Question: How important is the layerwise stacking and random initialization to making the ranVGG work? line 132. Eq4 w_l is scaled inversely to the magnitude of the gradient of the loss, what about making w_l scaled inversely to the magnitude of the loss instead?

Confidence in this Review

2-Confident (read it all; understood it all reasonably well)


Reviewer 2

Summary

This paper offers a method for performing deep visualization, including image inversion, texture synthesis and style transfer using *untrained*, randomly weighted DNNs.

Qualitative Assessment

Technical quality: The approach is sound and the results are impressive. Novelty: While the idea of using random weights in neural representations is very common, the way it is done here, and, in particular, the quality of the results attained seems like a nice contribution. Impact: The ability to perform these kinds of tasks using untrained models, and the ability to analyze network structures independently of (trained) weights is likely to be broadly useful, mostly from an efficiency point of view, but possibly even from a theoretical one as well. Presentation: Overall, the paper is well-organized and well-written, the exposition and pedagogy are very clear, the approach is appropriately formalized while remaining understandable---I enjoyed reading this paper. There are a few typos, grammar/usage errors, etc. that, while they don't seriously detract, could be corrected. Examples include: falmingo forth row (instead of fourth row in a couple of places) "well synthesis the textures of the original images" and a couple of other similar phrases. Also, from Fig 3 caption, it is not clear what is meant by the phrase, "Illustrations from the 10th L-BFGS iteration". Finally, the Van Gogh picture used in the paper is actually titled, "Starry Night", not "Night Starry"; also, it's date should be 1889, not 1989. Summary: This is a nice paper that takes an idea that is one of those obvious-once-you-show-me kind of things and makes some nice headway with it, demonstrates some good results and suggests several interesting avenues that will likely be explored by the NIPS community. It will make a good contribution to the conference.

Confidence in this Review

2-Confident (read it all; understood it all reasonably well)


Reviewer 3

Summary

The authors explore if an untrained CNN can be used as a generative model. They use some automatic normalisation and scaling of losses to make this work and show that they can indeed.

Qualitative Assessment

See above.

Confidence in this Review

2-Confident (read it all; understood it all reasonably well)


Reviewer 4

Summary

This paper shows that several recent visualization and style transfer techniques can be done with surprisingly high quality using only untrained (randomly initialized only) networks. It shows this on code inversion, texture synthesis and style transfer. It also provides some scaling considerations that are important to make it work with random weights (Gatys et al. have previously reported that this does not work, so these details seem to be important).

Qualitative Assessment

I think this is a very interesting finding, especially considering that this was claimed not to be possible in Gatys et al. (Texture Synthesis Using Convolutional Neural Networks). Your paper makes a comment about how your improved scaling of the gradients for the different layers is the difference maker. However, it would be interesting to see further study into this with experiments showing exactly how the results get worse as the scaling is artificially made less appropriate. I also think these results could have impact on application. Training time is one thing, but I think it is ever more useful not to have to store the weights. This can have significant impact for mobile apps, especially considering that results on these things tend to be much better on larger networks as opposed to simpler (Gatys et al.). You are also in a good position to do further investigation into this. For instance, we have seen examples of using various depth, but how do results vary with the number of parameters? How do results vary if you change the bias? I think overall trained networks still exhibit a little bit stronger results than untrained, even though it is pretty close. To me, the big open question however is if this is due to the supervised training or simply because of different statistics of the filters. If this could be understood, better sampling strategies for the convolutional layers might bridge this gap. Specific comments: - In (3), the weight factor w_l is missing. - About the initialization using sigma=0.015: So you do not re-normalize the network like Gatys et al.? Because a re-normalization process would effectively make the initial sigma irrelevant. Why not keep it simple and draw from N(0, 1) and then re-normalize? This would be similar in spirit to Xavier initialization, which is another option here. - The results in Fig. 5 for Gatys et al. look a bit weaker than in the original paper. What is the reason for this or am I just imagining things? - You might want to cite works like "On Random Weights and Unsupervised Feature Learning" by Saxe et al. This paper and a few more that it cites are a bit dated and deal with much more shallow networks than current models. However, it could still be interesting to add this to the discussion for a more complete overview of using untrained networks.

Confidence in this Review

3-Expert (read the paper in detail, know the area, quite certain of my opinion)


Reviewer 5

Summary

This paper performs 3 optimization tasks that are popular in deep visualization: 1) feature inversion, 2) texture synthesizing and 3) style transfer via a random network instead of a well trained network as often done in the literature. The result shows that a random network can be used to perform these 3 tasks well, and hints that random networks can be used to perform comparative analysis between different net architectures.

Qualitative Assessment

OVERALL: This is an interesting paper and I enjoy looking at the resultant images. The paper does raise an interesting idea that one could use random nets for inverting representation in order to understand architectures; but, did not actually show any result to back up that idea. The texture synthesis and style transfer experiments with a random net demonstrate a new and more efficient way (without having to train a net) to perform these tasks. However, the paper merely shows the results, but does not offer thorough investigation and insights into using random nets. It'd be great to have some of these questions answered in the paper: 1. is it surprising that inverting a random function/network works here? 2. how easy it is to invert a random nets compared to a trained net? 3. what are the disadvantages of using a random vs trained net? 4. how would the results change if performing these tasks with other architectures, e.g. (shallow) AlexNet, and (very deep) ResNet? TECHNICAL QUALITY 1. Inverting deep representations. - For a fairer comparison (qualitatively and quantitatively) with Mahendran et al, one should use the same setup (i.e. AlexNet and/or regularizer). Why was VGG used here? - Inverting from layer1-5 has been shown to be easy (Mahendran et al 2014, Dosovitskiy & Brox 2016). I wonder if random weights would still work similarly for higher layers? - This section would be much more impactful if the authors indeed inverted and compared different net architectures. - One way to improve the inversion quality further is to optimize an image to match both the code and the texture (Gram matrix) of a target image. 2. Textures synthesis: the results looks qualitatively great, and interesting as it's in contrast with the finding by Gatsy et al. - I wonder if it would produce a large variation (i.e. does starting from different random images produce very different samples) 3. Style transfer: Fig. 5 shows a recognizable artifact (rough, dark blobs of colors) in ranVGG compared to [13] which is smoother / qualitatively better. It'd be great to show more investigation into why it happened and see if the trained weights matter a lot here or it's just due to different hyperparam settings. NOVELTY Inverting and/or harnessing deep representations extracted from random networks has been previously studied in these two papers: - Alexey Dosovitskiy and Thomas Brox. Inverting visual representations with convolutional networks. arXiv preprint arXiv:1506.02753, 2015. - Alexey Dosovitskiy and Thomas Brox. Generating images with perceptual similarity metrics based on deep networks. arXiv preprint arXiv:1602.02644, 2016. Their approach is not the same as the one in this paper. But the general idea is the same, thus, their works in using random nets should be mentioned. Using random nets in texture synthesis and style transfer is novel/original, afaik. There is this concurrent texture synthesis work with similar ideas though: https://arxiv.org/pdf/1505.07376v3.pdf. POTENTIAL IMPACT The texture synthesis result shows high practical impact to the community. However, the representation inversion result is not very surprising and the authors have not demonstrated how such a random net can be used to understand different net architectures (as stated in the Intro & Discussion). CLARITY & PRESENTATION Overall, the paper is nicely written and easy to read. I see a few places that could be improved for clarity and presentation: L61-63: "Could we do deep visualization using untrained nets": the authors did not describe the scope of "deep visualization", but if it includes synthesizing a preferred images for neurons (see Nguyen et al 2016), then a random net wouldn't work. L16-17: The authors state that this proposed method sheds light on new understandings of deep vis, but the paper did not really discuss this in depth. L56-57: The purpose of a visualization technique is to enhance some understanding, thus I'm not sure what it's meant here.

Confidence in this Review

3-Expert (read the paper in detail, know the area, quite certain of my opinion)


Reviewer 6

Summary

The paper doubts the necessity of using trained CNN in order to perform different deep visualizations such as texture synthesis, style transfer and "inverting representations". It claims that the reasons that led to success in previous works should be contributed to the architecture itself and not the learned weights. The paper suggests a gradient based method to do such tasks while using un-trained networks.

Qualitative Assessment

1) inverting deep representations: the authors mention a greedy approach to build a stacked random weight network where for each layer they fix the weights of the lower layers, then they sample several sets of random weights for that layer and reconstruct the target image by choosing the weights yielding the smallest loss. I don't understand how the outcome of this process is considered a random network. The weights are manipulated all along the way. 2) If the main claim is that training is not needed then it would be also interesting to see a reproduction of previous work when using un-trained networks 3) line 123: what is pre_image? 4) line 202: (cargo at the second column). where? what figure? 5) Overall the results are not evaluated thoroughly. 6) Figure 3: Not clear how Euclidean distance is measured. Also not sure that Euclidean distance is the correct metric to be used here. Perhaps L infinity?

Confidence in this Review

1-Less confident (might not have understood significant parts)